# The Role of Artificial Intelligence in Monitoring Inflammatory Bowel Disease—The Future Is Now

**DOI:** 10.3390/diagnostics13040735

**Published:** 2023-02-15

**Authors:** Claudia Diaconu, Monica State, Mihaela Birligea, Madalina Ifrim, Georgiana Bajdechi, Teodora Georgescu, Bogdan Mateescu, Theodor Voiosu

**Affiliations:** 1Gastroenterology Department, Colentina Clinical Hospital, 020125 Bucharest, Romania; 2Internal Medicine Department, Carol Davila University of Medicine and Pharmacy, 50474 Bucharest, Romania

**Keywords:** artificial intelligence, automated diagnosis, inflammatory bowel disease, Crohn’s disease, ulcerative colitis, IBD-associated neoplasia

## Abstract

Crohn’s disease and ulcerative colitis remain debilitating disorders, characterized by progressive bowel damage and possible lethal complications. The growing number of applications for artificial intelligence in gastrointestinal endoscopy has already shown great potential, especially in the field of neoplastic and pre-neoplastic lesion detection and characterization, and is currently under evaluation in the field of inflammatory bowel disease management. The application of artificial intelligence in inflammatory bowel diseases can range from genomic dataset analysis and risk prediction model construction to the disease grading severity and assessment of the response to treatment using machine learning. We aimed to assess the current and future role of artificial intelligence in assessing the key outcomes in inflammatory bowel disease patients: endoscopic activity, mucosal healing, response to treatment, and neoplasia surveillance.

## 1. Introduction

Inflammatory bowel diseases (IBD), such as ulcerative colitis (UC) and Crohn’s disease (CD), pose significant challenges with regard to diagnostic and management strategies. Moreover, their incidence is increasing globally and their impact on one’s quality of life is not negligible [1]. Important heterogeneity exists in both the quality and interpretation of diagnostic information due to differences in physician experience and clinical practice, leading to a considerable variation in disease management among gastroenterologists [2].

One of the most important complications of IBD is its association with an increased rate of colorectal neoplasia. Dysplasia and colorectal adenocarcinoma are associated with longstanding active inflammation, severe disease, colonic stricture, and post-inflammatory polyps, as well as other risk factors that are not directly linked to IBD, such as a personal history of dysplasia, a family history of colorectal cancer, and concomitant primary sclerosing cholangitis [3]. Endoscopy remains the cornerstone of IBD management, with a key role in diagnosis, treatment, and surveillance. AI could further enhance the role of endoscopy, from grading the disease severity, the assessment of the response to treatment, or neoplasia surveillance [4]. We aimed to review the current and future role of artificial intelligence (AI) in assessing the critical key outcomes in IBD patients: disease activity, mucosal healing, response to treatment, and neoplasia surveillance.

## 2. Understanding the Role of Artificial Intelligence in Gastroenterology

Advances in AI are driving important changes in medicine, and it is expected to provide, in the near future, significant improvements in patient care across a wide range of clinical settings [5]. AI has the ability to analyze large amounts of complex data at a significantly faster pace than humans, highlighting details that might be overlooked by the human eye, ensuring a precise and objective evaluation of the data [6]. AI applications include machine learning, neural networks, and deep learning (Figure 1). The fundamental principle is machine learning (ML), which is defined as the ability to automatically build mathematical algorithms from the input of raw training data in order to make decisions in new circumstances without human surveillance [7]. They can learn from experience, without being specifically programmed.

Deep learning (DL) is a fast-growing machine-learning method which has become the dominant approach for recent work in the ML field in recent years. Convolutional neural networks (CNNs), inspired by the neural network of the human brain, can enable a fast and accurate image discrimination and video analysis [8]. These applications of AI can be used in upper gastrointestinal endoscopy, such as the assessment of early gastric cancer, the identification of H. pylori, dysplasia in Barrett’s esophagus or colonoscopy, for colorectal polyp detection, and for assessing advanced neoplasia in colonic polyp or endocytoscopy to predict persistent histological inflammation in inflammatory bowel disease, allowing for breakthroughs in medical imaging recognition [9,10].

Computer-aided diagnosis (CAD) systems have been recently introduced in clinical practice (EndoBrain, GI Genius, Discovery, Endo-Aid, CAD EYE, Endo-screener or Wise vision.), providing the real-time detection and diagnosis of endoscopic lesions, acting as a quality controller and training vector for endoscopists [11]. The main advantages offered by CAD systems compared to traditional imaging methods are a more comprehensive imaging information compiling, better reproducibility, and the ability to implement an automatic selection of the region of interest.

CAD EYE^TM^ (Fujifilm, Tokyo, Japan) is the first CAD system to combine computer-aided detection (CADe), which detects gastrointestinal lesions, and computer-aided diagnosis (CADx), which characterizes gastrointestinal lesions on the same platform (Figure 2), demonstrating a better performance than the human eye [12]. CADe uses LCI to enlighten differences in color in the red zone and CADx uses BLI, which varies the light emission ratios of multiple lights with different wavelengths to distinguish polyps by intensifying minute vessels and structures in the mucosa [13]. A retrospective trial of colorectal polyps evaluated its effectiveness by using endoscopic images obtained from seven centers as validation images. The detection sensitivities of white light imaging (WLI) and linked color imaging (LCI) for the CADe system were 94.5% and 96%. The accuracy of WLI and blue light imaging (BLI) in CADx was 93.2% and 94.9% [12]. However, to date, the CAD EYE system can only be used to evaluate colorectal lesions, which it can only classify as neoplastic or hyperplastic, with further applications currently under development (such as diagnosis of cancer invasion depth, prediction of metastasis, and recurrence) [14].

## 3. Potential Applications of AI in IBD

In IBD, the endoscopic assessment of disease extension and severity, as well as mucosal healing and the early detection of neoplasia, represent key factors in ensuring adequate patient management [15,16]. Emerging as a valuable tool in IBD diagnosis and management, artificial intelligence offers the possibility of the simultaneous analysis of miscellaneous biological data by permitting a large-volume input for machine learning models, such as cross-sectional imaging, endoscopic and histologic imaging, inflammation biomarkers, as well as gut microbiota composition and gene expression [17,18,19]. ML algorithms can learn relevant features from existing patient databases and compare them to the known outcomes, which can, in turn, be used to predict the patient’s prognosis. As the application areas of AI in IBD will continue to expand, one significant area of interest is represented by the long term follow-up of these patients, including the prediction of the treatment response and relapse as well as screening for IBD-associated colonic neoplasia (Figure 3).

## 4. The Role of Artificial Intelligence in Assessing Disease Activity

An endoscopic assessment continues to represent the gold standard for diagnosis, disease severity assessment, and evaluating the response to therapy [20,21]. Efforts to standardize the endoscopic scoring of disease activity have resulted in many endoscopic scores [15,16]. However, the endoscopic assessment of disease severity is limited by the fact that endoscopic scores are rarely used in everyday practice and most scoring systems are still not yet validated [15], as well as the fact that this assessment is subjected to interobserver variability [22].

Advances in the use of artificial intelligence can offer a solution to the inherent subjectivity of human interpretation, aiming to eliminate bias and variability, as well as improving the precision and accuracy in quantifying disease severity. Moreover, integrating AI algorithms in an endoscopic assessment offers the possibility of analyzing large databases and identifying occult disease patterns. In recent years, a great body of evidence has emerged regarding the role of AI in assessing disease activity, as deep learning algorithms have shown satisfying results in UC [23,24,25,26]. Bhambhavi et al. have trained a CNN model using still images of endoscopy in order to recognize and classify images according to the endoscopic Mayo score (EMS); the final model classified MES 3 disease with an AUC of 0.96, MES 2 disease with an AUC of 0.86, and MES 1 disease with an AUC 0.89; the overall accuracy was 77.2% [23] Gutierrez et al. trained a deep learning-based system to assess the EMS on raw full length colonoscopies collected from Etrolizumab clinical trials, with great results being obtained (AUROC = 0.84 for Mayo Clinic Endoscopic Subscore ≥ 1, 0.85 for Mayo Clinic Endoscopic Subscore ≥ 2, and 0.85 for Mayo Clinic Endoscopic Subscore ≥ 3, respectively) [24]. In another study, Yao et al. obtained a 57.1% automated and central reviewer agreement, which improved to 69.5% when the reviewer disagreement was taken into consideration [26]. Last but not least, Takenaka et al. developed a deep neural network algorithm for UC by using almost 40,000 images of colonoscopy and almost 7000 biopsy results from 2012 patients with UC undergoing a colonoscopy between January 2014 and March 2018, while patients who underwent a colonoscopy between April 2018 and April 2019 were used for validation purposes; endoscopic remission was defined as the UC endoscopic index for a severity score of 0, while histologic remission was defined as a Geboes score of 3 or less. The algorithm identified patients with endoscopic remission with a 90.1% accuracy (95% confidence interval [CI] 89.2–90.9%) [23]. Although not as significant, data on the impact of AI in CD patients is emerging as well [27,28,29]. Table 1 summarizes the role of AI in IBD assessment and management.

Treatment targets have shifted significantly in the past decade, from achieving clinical remission to mucosal healing (MH), an endpoint that could alter the natural disease course [30]. MH is associated with a lower risk of relapse, hospitalization, surgery, and neoplasia [31]. There is no consensus regarding the definition of MH, despite the many endoscopic scoring systems proposed in recent years. Recent society guidelines [32] state that the following criteria are considered acceptable criteria of mucosal healing: for Crohn’s disease, a Simple Endoscopic Score for Crohn’s Disease (SES-CD) of <3 or the absence of ulcerations (SES-CD ulceration subscore = 0), and for ulcerative colitis, a Mayo endoscopic subscore = 0 points or UCEIS ≤1. However, these scores lack validation in prospective studies and reproducibility is substandard. For example, a study conducted by Daperno et al. [33] has shown that for both the Mayo endoscopic subscore and SES-CD, the inter-observer agreement is suboptimal. 

The results of a recent systematic review showed that AI algorithms for the prediction of endoscopic or histologic disease activity in UC performed with an overall sensitivity and specificity of 78% (median, range 72–83, IQR 5.5) and 91% (median, range 86–96, IQR 5), respectively [34].

Ozawa et al. [35] evaluated 841 colonoscopies from patients diagnosed with ulcerative colitis with the help of a trained convolutional neural network (CNN) and observed a high level of performance in identifying disease activity of Mayo 0 and 0–1 (AUC 0.86 and 0.98, respectively), with a better performance in recognizing Mayo 0 scores in the rectum compared to the left or right colon (AUC = 0.92, 0.83 and 0.83, respectively). Similarly, Takenaka et al. [36] trained a deep neural network (DNN) to assess endoscopic and histologic disease activity based on the ulcerative colitis endoscopic index of severity score (UCEIS) and the Geboes score of histology, defining MH as the combination of an UCEIS score of 0 and a Geboes’ score of ≤3. The authors found that the deep neural network had a high sensitivity (92.0%) and specificity (91.3%) for evaluating mucosal healing. Correlation coefficients between the DNN and expert endoscopists and pathologists were also remarkable (0.917 and 0.859, respectively), showing that DNN may have better overall results when compared to CNN. It is worth mentioning that data were gathered from a single tertiary care center for IBD and, as such, their applicability in a wider clinical practice might not be as successful. These results were later applied in a CAD-driven endoscopic assessment of UC by the same study group, showing potential in predicting patient prognosis [25].

**Table 1 diagnostics-13-00735-t001:** Summary of existing studies using artificial intelligence in IBD management.

Study	Study Type	Modality	AI Classifier	Aim of AI Use	Training Set	Results		
						Accuracy	Sensitivity	Specificity
Quénéhervé L et al. [37]	Retrospective	CFLC	Automated analysis method	Discrimination between CD and UC	12.900 images	91.0	74.0	97.0
Stidham R et al. [38]	Retrospective	WLI	CNN	Discriminating endoscopic remission from moderately-severe disease UC	16.514 images		83.0	96.0
Bossuyt P et al. [39]	Prospective	WLI	Integration of pixel color data	Assessment of disease activity in UC	35 patients	R = 0.65 RD correlated with RHI		
Ozawa T et al. [35]	Retrospective	WLI	CNN	Mucosal healing in UC	−26.304 images	AUROC 0.98 (Mayo 0–1)		
Takenaka et al. [36]	Prospective	WLI	DNN		40.758 images	90.1	93.3	87.8
Maeda et al. [40]	Retrospective	EC	SVM	Prediction of persistent inflammation	12.900 images	91.0	74.0	97.0

AI, artificial intelligence; IBD, inflammatory bowel disease; EC, endocytoscopy; CLEC, confocal laser endomicroscopy; WLI, while light imaging; CD, Crohn’s disease; UC, ulcerative colitis; SVM, support vector machine; CNN, convolutional neural network; AUROC, area under the receiver operating curve; RD, red density; RHI, Robarts histological index.

Histologic healing is not yet included in the therapeutic targets in IBD management but is gaining increased attention and could be used as an adjunct to endoscopic remission to represent a deeper level of healing [21]. Some authors started researching deep learning strategies focused on histologic healing. Gui et al. [41] recently developed a histological index, aligned to endoscopy and suited to apply to an AI system to evaluate the inflammatory activity, the Paddington International virtual ChromoendoScopy ScOre (PICaSSO) Histologic Remission Index (PHRI). This AI algorithm differentiated active from quiescent UC with a 78% sensitivity, 91.7% specificity, and 86% accuracy.

As the persistence of histological inflammation is a risk factor for clinical relapse as well as a driver for colorectal neoplasia, AI-assisted methods of detecting residual inflammation can be an important tool in long-term surveillance in IBD populations. Maeda et al. [41] used a CAD-assisted machine learning model to detect the severity of histologic inflammation using endocytoscopy-enhanced colonoscopy still images from ulcerative colitis patients, achieving an accuracy, sensitivity, and specificity of 90%, 74%, and 91%, respectively. However, the applicability of endocytoscopy-enhanced colonoscopy is limited in real life practice due to increased costs and procedural times as well as the limited number of physicians with EC experience.

The versatility of AI means that algorithms are not only limited to conventional endoscopy but can also be applied to videocapsule endoscopy (VCE) [42]. In a study of AI in Crohn’s disease patients undergoing videocapsule endoscopy, Barash et al. [43] observed an accuracy of the algorithm of 0.91 for discriminating grade 1 vs. grade 3 ulcers, 0.78 for grade 2 vs. grade 3, and 0.624 for grade 1 vs. grade 2.

Even though colonoscopy with ileal intubation is the first-line investigation in suspected IBD, VCE plays a pivotal role [15,44], along with other imaging modalities such as enteroscopy and cross-sectional imaging in diagnosing CD restricted to the small bowel. Adding AI to conventional VCE could increase the diagnosis accuracy and help characterize disease extend and severity. AI could enhance the detection rate of subtle ulcers that are difficult to discriminate from normal tissue. Fan et al. were the first to exploit a deep learning framework on automated ulcer and erosion detection in VCE images with promising results: the accuracy for the ulcer was 95.16% and 95.34%, a sensitivity of 96.80% and 93.67% was obtained, and a specificity of 94.79% and 95.98% was achieved, correspondingly [45]. Another study showed that a CNN system reduced the reading time of endoscopists without decreasing the detection rate of mucosal breaks (3.1 vs. 12.2 min) [46].

## 5. The Role of Artificial Intelligence in Screening for Early Neoplasia in IBD

The association of longstanding inflammatory bowel diseases (IBDs), especially ulcerative colitis, with colorectal cancer is already well acknowledged. A young age at diagnosis, longer disease duration, higher inflammatory burden, greater extent, family history of colorectal cancer (CRC), and association with primary sclerosing cholangitis are the risk factors for neoplasia development [47,48]. Persistent levels of inflammation, with repeated flares of disease, can lead to the oncogenic insult of the colonic epithelium in these patients [49,50].

Colonoscopic findings in IBD surveillance can be classified as polypoid or nonpolypoid lesions and invisible dysplasia. Sporadic adenomas may appear as discrete, visible lesions, but they also appear as a “field cancerization” that develops in IBD when the entire mucosa is chronically inflamed, increasing the risk of synchronous and metachronous neoplasms [51]. The current guidelines (European Crohn’s and Colitis Organisation—ECCO, American Gastroenterological Association—AGA, and the British Society of Gastroenterology—BSG) recommend that surveillance colonoscopies should begin in 8–10 years after the onset of the symptoms, and should be done at 1, 2–3, and 5 years in high-, intermediate-, and low-risk patients, respectively [52,53,54,55,56]. Patients with colonic stenosis detected within 5 years after diagnosis should have a low threshold for cancer screening, as they are at a high risk of developing CRC and a colonoscopy should be performed annually [57].

Current surveillance strategies include high-definition endoscopy and chromoendoscopy, with indigo-carmine or methylene blue, and targeted biopsies of abnormal appearing mucosa [58]. Virtual chromoendoscopy is considered a suitable alternative to dye chromoendoscopy when using high-definition endoscopy [59,60]. If virtual or dye-based chromoendoscopy are not available, non-targeted biopsies every 10 cm should be taken and additional biopsies should be collected from areas of previously known dysplasia or poor mucosal visibility.

One meta-analysis [58] revealed that chromoendoscopy increases the yield of dysplasia compared with white-light endoscopy (absolute risk increase = 6% (3–9%)). However, conventional chromoendoscopy is a time-consuming and operator-dependent method, requiring an adequate bowel preparation and mucosal healing [61].

Despite the development of high-definition endoscopes and dye-based chromoendoscopy, the mortality and morbidity related to IBD neoplasia remains high [62,63]. In order to address some of the limitations in the current strategies of neoplasia surveillance, such as a high variability in disease presentation and the associated risk, imperfect endoscopic techniques, or a high susceptibility to interobserver variability in lesion assessment, artificial intelligence was explored to aid traditional colonoscopy [64]. Many AI algorithms were developed in order to alert the endoscopist of neoplastic lesions in real-time by using visual and auditory signals during the colonoscopy [65].

With this purpose, CNN were trained to detect neoplastic lesions in the non-IBD population, using still images annotated by expert endoscopists, proving a good sensitivity and specificity for lesion detection. Hassan et al. showed that the GI-Genius Medtronic system reached a sensibility of 99.7% in polyps’ detection [66], while another recent computer-aided detection system demonstrated an increased sensitivity for all, diminutive, protruded, and flat polyps (98%, 98.3% and 97%, respectively) [67]. However, its use for the detection of dysplasia in patients with IBD has not been concluded. Fukunag reported the case of a high-grade dysplasia flat lesion detected by the EndoBRAIN system in a patient with longstanding colitis [68], which was successfully removed via submucosal dissection.

In IBD patients, CADe/CADx systems are useful in the detection and differentiation of colon polyps/lesions and for dysplasia surveillance [4]. Additionally, virtual chromoendoscopy (VCE) was recently evaluated for the potential role of the identification of dysplastic lesions and it seemed to have a similar detection rate of dysplasia in IBD as high-definition WLE (HD-WLE) [69]. This study showed similar intraepithelial neoplasia detection rates between VCE-enhanced colonoscopy with targeted biopsies and WL colonoscopy with targeted and stepwise random biopsies (57% vs. 48%, respectively), but with significantly longer procedure times and higher numbers of acquired biopsy specimens in the WLE group. There are still challenges in clinical practice for IBD patients in using AI technology, such as the ability to differentiate pseudopolyps from true polyps and to detect flat lesions which appear mostly in long-standing colitis [70].

## 6. AI in Aiding IBD Treatment—Disease Progression Prediction/Response to Treatment

In more recent years, the goal of IBD treatment have evolved from traditional clinical remission to a more integrated and complete mucosal healing and deep remission [71,72,73].

Despite ongoing development in IBD therapies, with newer drugs ranging from biologics that interfere with the inflammatory cascade (anti-tumor necrosis factor-α, anti-interleukin-12/23, anti-integrins) to small molecules (JAK-inhibitors) [74,75], clinicians still lack the adequate tools for predicting the treatment response, thus adequately matching patients and drugs, thereby improving the patient outcomes and reducing the financial burden of these treatments [76]. Since the concept of artificial intelligence was popularized, its applicability in disease progression and treatment response prediction has become a major subject of interest. Researchers have used random forest (RF) classifiers on data gathered from hospital databases in order to predict the response to therapy [77]. Waljee and colleagues have conducted many studies in this domain [78,79,80,81,82,83]. In one of the first studies, they attempted to identify three different outcomes in patients treated with thiopurines (clinical response, thiopurine non-adherence, and patients who were most likely to shunt from 6-thioguanine nucleotide [6-TGN] to 6-methylmercaptopurine [6-MMP] metabolites). The models were efficient in predicting the outcomes, with an AUC of 0.86 [95% CI 0.79–0.92] for the clinical response [78]. In a more recent study, they have developed an algorithm using the same cohort and similar outcomes, except they focused on the objective response (defined as absence of intestinal inflammation) with an AUC of 0.79 (95% CI 0.78–0.81). Some of the most important variables included: hemoglobin, lymphocytes, hematocrit, neutrophils, and platelets [79].

Based on the data collected from large clinical trials, prediction models regarding the response to biological treatment (particularly to vedolizumab and ustekinumab) were evaluated [80,81,82,83]. Vedolizumab is a gut selective alpha-4-beta-7 integrin therapy approved for the treatment of ulcerative colitis (UC) as well as Crohn’s disease (CD) [80]. Waljee et al. have used three different RF models (baseline, week 6, and simplified) in order to predict corticosteroid-free Vedolizumab remission in CD patients (defined as no corticosteroid use and CRP reduction to ≤5 mg/dL) at week 52. Of these three, the Week 6 model and the simplified week 6 model (HGB * ALB * VDZ level)/(CRP * weight in kg) had the best accuracy (AUC 0.75; 95% CI 0.64–0.86 and AUC 0.75; 95% CI 0.70–0.81, respectively). Some of the most important variables used for the week 6 model were: CRP, slope of Vedolizumab level, hemoglobin, albumin, the vedolizumab level, and slope of CRP. Patients predicted to be in corticosteroid-free remission by the week 6 model achieved the endpoint in almost one third of cases (35.8%), while those predicted to fail succeeded in 6.7% of cases, therefore allowing the user to identify the majority of patients that are unlikely to achieve remission by week 6 [81].

In another study, the same author used baseline data and week 6 data from patients with UC treated with Vedolizumab (GEMINI II) and developed two models to predict corticosteroid-free remission at week 52, defined as no corticosteroid use and an endoscopic Mayo subscore of 0 or 1. A simplified week 6 model was also created, using a fecal calprotectin cut-off of under 234 μg/g to predict the composite outcome. However, the week 6 model proved to have a higher accuracy, with an AUC of 0.73 [95% CI: 0.65–0.82] [82].

Last but not least, the most recent study conducted by Waljee and colleagues focuses on predicting the biological remission at week 42 (defined as the CRP level under 5 mg/dL) for CD patients treated with ustekinumab in UNITI and IM-UNITI studies. They developed two models: one baseline and one for week 8, and also a simplified version, week 6 albumin-to-CRP ratio. The AUC for the week 8 model was 0.78 (95% CI, 0.69–0.87), with a similar value for the simplified model (0.76 (95% CI, 0.71–0.82)); the baseline levels of Ustekinumab did not improve the performance of the prediction model [83].

## 7. Discussion

The management of IBD patients remains a challenge to healthcare professionals as patients undergo the complex process of diagnosis, the evaluation of the disease severity, the response to treatment, the long-term follow-up, and complication management. The ability of AI to integrate large volumes of data theoretically would allow for a more targeted approach tailored to the patient’s disease subtype, concomitant comorbidities, and differences in the socioeconomic and psychological factors [84].

The application of AI has the potential to improve the accuracy and precision of predicting outcomes with various IBD treatments, but it is momentarily limited to a research setting.

There is a significant heterogeneity in the treatment response to biologics, therefore being able to predict the response after a short course is of utmost importance at an individual but also systemic level. If implemented in a real-life setting, AI-based management algorithms could help guide therapy and consequently reduce the costs associated with an expensive but unsuccessful treatment, as well as the complications associated with the suboptimal control of the disease.

Although exciting, the applications of AI in the management of IBD need to be considered in the context of their inherent challenges and limitations. Inherent pitfalls of AI consist of selection bias, spectrum bias [85], and the low complexity of algorithm development, which can lead to the inappropriate generalization of the results.

Current evidence is limited by the predominance of retrospective data on which the training of AI algorithms was performed [23,24,25,26,27,28,29,66,67,68]. Data retrieved from the cohorts enrolled in the clinical trials of investigational drugs is probably significantly different from a real-life setting. Additionally, CNN models trained on single-center databases had a poor performance when assessed for achieving a wide applicability, stressing the need for multi-center data acquisition and the external validation of AI algorithms [86]. Additionally, prospective studies using AI in IBD management are necessary to evaluate their efficacy, in conformity to the new CONSORT-AI and SPIRIT guidelines [87,88]. AI-assisted colonoscopies are also limited by the use of still images in algorithm development, which can be hard to adapt to real-life video colonoscopies. The challenge AI faces is analyzing raw full-motion videos and differentiating informative from non-informative frames (i.e., affected mucosa vs. unaffected mucosa) and applying different layers of analysis in real time, in order to offer an output on the disease type, severity, treatment response, or neoplasia development.

It is also important to clearly mention that AI models provide possibilities of a prediction and not an absolute answer. Moreover, clarifications should be made on the ethical implications of using potentially biased AI machines as well as the legal implications of erroneous decisions being made while using AI [89].

In an effort to overcome these limitations, a close collaboration between physicians, statisticians, and bioinformaticians is needed in order to develop algorithms capable of delivering clinically meaningful outputs [64].

## 8. Future Directions

The promise of AI in medicine is the ability to analyze and integrate a variety of data and provide information in innovative applications. The use of AI in gastroenterology in general, and in IBD management in particular, continues to evolve rapidly at multiple levels of patient care, enabling more than ever the possibility of “precision medicine”. The integration of monitoring devices (e.g., smartwatches and smartphones) that could send real-life data on the treatment response (such as the clinical data sent from the patients) could be integrated in the ML algorithm, which could receive much attention in the future. Prospective, multidisciplinary, and multicenter AI studies are needed to establish clinical use in monitoring IBD patients.

## 9. Conclusions

AI technology in IBD is still in a research phase that can only be experimentally used. However, the AI applications developed so far in the IBD field already have the potential to improve the standards for patient care, starting from the diagnosis to long-term therapeutic decisions and neoplasia surveillance.

## Figures and Tables

**Figure 1 diagnostics-13-00735-f001:**
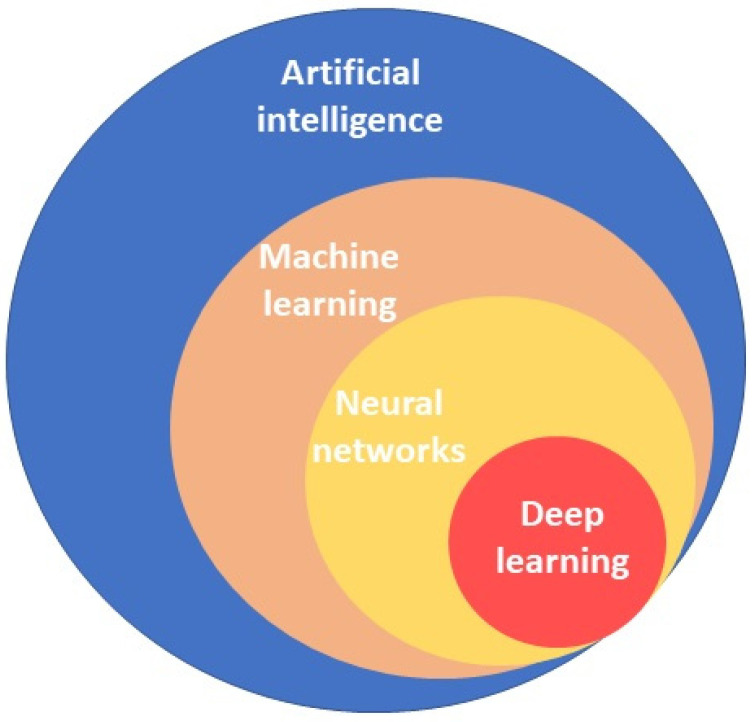
Primary concepts of artificial intelligence.

**Figure 2 diagnostics-13-00735-f002:**
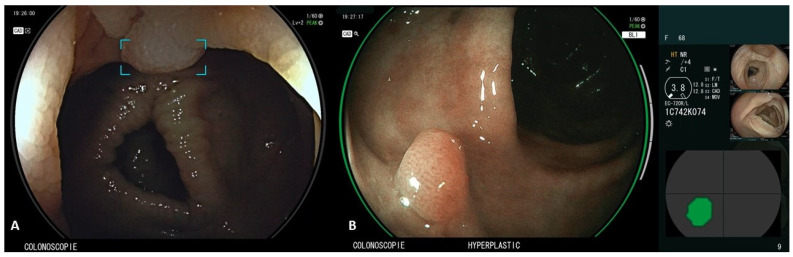
CAD EYE^TM^ Software (Fujifilm, Tokyo, Japan); panel (**A**)—CADe function. A polyp is identified in WLI and marked using an annotation box; panel (**B**)—CADx function. The polyp is evaluated by CAD EYE in BLI and a diagnosis of hyperplastic polyp is indicated at the bottom of the monitor screen. The position and outline of the polyp is also indicated in the lower right part of the monitor.

**Figure 3 diagnostics-13-00735-f003:**
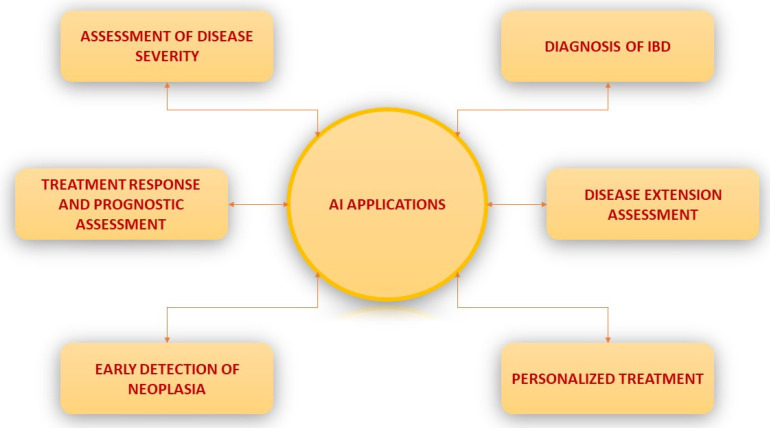
Graphical representation of potential application of AI in IBD.

## Data Availability

Not applicable.

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
