# Peer review of "The Role of Artificial Intelligence in Monitoring Inflammatory Bowel Disease—The Future Is Now"

_diagnostics, 2023, doi:10.3390/diagnostics13040735_

Round 1

Reviewer 1 Report

Final decision: Manuscript is acceptable for publication after minor correction.

Abstract: Current abstract is acceptable

Keywords: Current Keywords needs (could be) to be rewritten as: Artificial intelligence; automated diagnosis; inflammatory bowel disease; Using AI Template

Introduction: Acceptable

M & M: Current M & M is acceptable.

Results: Current results are acceptable.

Discussion: Acceptable

Bibliography/References: Acceptable

I could not see running title.

Final decision: Manuscript is acceptable for publication after minor correction as recommended.

Author Response

Dear reviewer,

On behalf of all the authors, I would like to thank you for you comments and suggestions. We addressed all the issues raised as follows:

  • We added a running title – “Monitoring IBD using AI”
  • Keywords were rewritten

Reviewer 2 Report

The paper is well written with a comprehensive review on this new field of research in IBD. There are no particular observation to be made for me and the paper  is ready for publication and will be very interesting for the reader.

Author Response

Dear reviewer,

On behalf of all the authors, I would like to thank you for you comment.

Reviewer 3 Report

1.It is suggested to add new related in vitro studies and compare with in silico methods.
2.what is the suggestion of this study for future works?
3.Please discuss and compare your results with clinical cases.
4.It will be better to add the role of mitochondria in monitoring.
5.Please add details for time period and dose selection.

Author Response

Dear reviewer,

On behalf of all the authors, I would like to thank you for you comments. We contacted the editorial office to make sure your uploaded response was meant for our manuscript, as some of the comments do not refer to its content. We are currently waiting for further instructions.